# Genome Sequencing of *Lentinula edodes* Revealed a Genomic Variant Block Associated with a Thermo-Tolerant Trait in Fruit Body Formation

**DOI:** 10.3390/jof10090628

**Published:** 2024-09-02

**Authors:** Seung-il Yoo, Suyun Moon, Chang Pyo Hong, Sin-Gi Park, Donghwan Shim, Hojin Ryu

**Affiliations:** 1Division of Bioinformatics, Invites Biocore, Seoul 08511, Republic of Korea; seungil.yoo@inviteseco.com (S.-i.Y.); singi.park@inviteseco.com (S.-G.P.); 2Department of Biology, Chungbuk National University, Cheongju 28644, Republic of Korea; sooym21@gmail.com; 3Department of Crop Science and Biotechnology, General Graduate School, Dankook University, Cheonan 31116, Republic of Korea; changpyo.hong@dankook.ac.kr; 4Department of Biological Sciences, Chungnam National University, Daejeon 34134, Republic of Korea; 5Center for Genome Engineering, Institute for Basic Science, Daejeon 34126, Republic of Korea

**Keywords:** *Lentinula edodes*, comparative genomics, heat stress, domestication, CAPS

## Abstract

The formation of multicellular fruiting bodies in basidiomycete mushrooms is a crucial developmental process for sexual reproduction and subsequent spore development. Temperature is one of the most critical factors influencing the phase transition for mushroom reproduction. During the domestication of mushrooms, traits related to fruiting bodies have significantly impacted agricultural adaptation and human preferences. Recent research has demonstrated that chromosomal variations, such as structural variants (SVs) and variant blocks (VBs), play crucial roles in agronomic traits and evolutionary processes. However, the lack of high-quality genomic information and important trait data have hindered comprehensive identification and characterization in *Lentinula edodes* breeding processes. In this study, the genomes of two monokaryotic *L. edodes* strains, characterized by thermo-tolerance and thermo-sensitivity during fruiting body formation, were reassembled at the chromosomal level. Comparative genomic studies of four thermo-tolerant and thermo-sensitive monokaryotic *L. edodes* strains identified a 0.56 Mbp variant block on chromosome 9. Genes associated with DNA repair or cellular response to DNA damage stimulus were enriched in this variant block. Finally, we developed eight CAPS markers from the variant block to discriminate the thermo-tolerant traits in *L. edodes* cultivars. Our findings show that the identified variant block is highly correlated with the thermo-tolerant trait for fruiting body formation and that alleles present in this block may have been artificially selected during *L. edodes* domestication.

## 1. Introduction

*Lentinula edodes*, commonly known as ‘shiitake’ or ‘pyogo’, is one of the most popular and economically significant edible mushroom species. It is widely valued for its rich flavor and nutritional benefits and is broadly cultivated worldwide [1]. In general, pyogo mushrooms are divided into three cultivation types based on their optimal fruiting body formation at low, medium, and high temperatures [2]. During domestication and breeding, many pyogo cultivars have been selected based on their temperature spectral range for proper reproductive fruiting body formation. However, growing pyogo mushrooms is frequently hampered under high-temperature conditions, which can result in diminished mycelial growth and production [3]. High temperatures have the potential to disrupt biological processes, denature proteins, and alter numerous metabolic pathways [4]. Organisms that can tolerate high temperatures have evolved strategies and mechanisms to counteract these effects and maintain cellular homeostasis [5]. Understanding the genetic basis of thermotolerance in *L. edodes* is crucial for developing improved strains that can withstand high-temperature environments.

High temperatures during the summer season generally diminish the vitality of mycelium, facilitating the invasion of additional pathogens into the log or sawdust medium. This effect significantly contributes to the decline in both the yield and quality of edible mushrooms [1]. By identifying genes and genetic variations linked to high-temperature tolerance, we can enhance our understanding of the molecular mechanisms underlying this characteristic. This knowledge can be utilized to establish breeding programs aimed at enhancing thermo-tolerance in pyogo cultivation. Research has shown that heat stress can lead to oxidative damage, which can be mitigated by the action of para-aminobenzoic acid (PABA) synthase and nitric oxide, reducing the levels of reactive oxygen species (ROS) [6,7]. Several factors, such as catalase, trehalose, heat shock proteins (HSPs), and calcium–calmodulin signaling pathways, have been identified as significant modulators in the regulation of the heat stress response in various edible mushrooms [8]. A study demonstrated that the addition of external auxin can improve the ability of *L. edodes* mycelium to withstand high temperatures [9]. Additionally, the role of the anthranilate-synthase gene (TrpE) during heat stress in *L. edodes* has been investigated through knockdown and overexpression experiments [9,10]. This suggests that the thermotolerance of fungal species is strongly influenced by evolutionarily conserved genetic regulatory mechanisms. However, comprehensive studies specifically aimed at identifying and understanding the characteristics of different fungal species at the genome level are still lacking. Previous studies on *Lentinula edodes* genome assemblies have made substantial contributions to understanding the genetic architecture of this important mushroom species. For example, Chen et al. [11] reported a genome assembly with over 340 scaffolds, while Shim et al. [12] used PacBio sequencing to achieve a more contiguous assembly with 31 scaffolds. However, these assemblies still lacked chromosome-level resolution, limiting their utility for detailed genomic studies.

Assessing genetic variability among varieties is crucial not only for the protection of varietal rights but also for practical reasons such as conserving and expanding the genetic base of resources [13]. Various molecular markers can be used to assess genetic variation, and advances in sequencing technology (NGS) have made it quick and straightforward to detect sequence variations, including SNPs and indels, that support differences between alleles. NGS technology has led to significant advances in genomic studies by enhancing the amount and accuracy of sequence information and enriching genomic resources [13]. Additionally, NGS allows for the discovery of new or reference-based genomic variations, even in organisms with little or no genetic information [14,15]. These large-scale sequence data sets can also be used to define sequence diversity and create polymorphism and genotyping data. Furthermore, NGS is valuable for discovering, validating, and assessing genetic markers [15]. Resequencing individuals with diverse germplasms using NGS technology has become routine for discovering sequence variations. These strategies have been employed in crops such as rice, soybean, chickpea, and potato, and in fungi, including *Aspergillus carbonarius* and *Tuber melanosporum* [16,17,18].

Recent advances in functional genomics and genetic approaches have led to the discovery of intricate interplays between cellular signaling networks and environmental stimuli that play a crucial role in the life cycle of fungi and edible mushrooms [19]. Understanding the genetic basis of the thermotolerance characteristic in *Lentinula edodes* offers the potential to cultivate enhanced strains suitable for high-temperature conditions. Genetic investigations in this context can provide valuable insights into the molecular mechanisms underlying this trait, including the identification of genes and chromosomal regions that contribute to the response to high-temperature stress. This work aimed to enhance the genomic information on monokaryotic strains generated from pyogo cultivars Sanmaru1 and Sanjo502, which exhibit varying levels of tolerance to high temperatures during fruiting body formation. Through the application of comparative genomics on eight thermo-tolerant or -sensitive monokaryotic strains, a variation block (VB) of 0.56 Mbp was identified on chromosome 9, which is predicted to be involved in high-temperature tolerance. Within this VB, we found a significant enrichment of GO terms related to DNA damage stimulus and DNA repair. The development of the CAPS marker involved the utilization of seven single nucleotide polymorphisms (SNPs) located in the temperature-responsive variant block. These CAPS markers were then used to evaluate the discrimination of thermotolerant cultivars of pyogo.

## 2. Materials and Methods

### 2.1. Fungal Materials and Cultivation Conditions

Fungal materials used in this study were provided by the National Institute of Forest Science in the Korean Forest Service (http://www.forest.go.kr, accessed on 22 August 2024, Sanmaru1, 2 and Kinko135) and the Forest Mushroom Research Center (http://www.fmrc.or.kr, accessed on 22 August 2024, Sanjo 501 and 502). The mycelia of the strains were maintained on potato dextrose agar (PDA) under dark conditions at 25 °C. To analyze the phenotype of high-temperature tolerance during fruiting body formation, each strain was cultured on sterilized oak sawdust medium (4:1 oak sawdust and rice bran with a water content of approximately 65%). The oak sawdust medium was sterilized at 105 °C for 10 h before spawn inoculation. The inoculated sawdust media were incubated in the dark for 40 days and then exposed to light for 60 days. The phenotype of fruiting under high temperatures was analyzed by inducing fruiting at 25 °C.

### 2.2. Genomic DNA Extraction and Whole-Genome Sequencing

The genomic DNA of Sanmaru and Sanjo was extracted using the Wizard^®^ HMW DNA Extraction Kit (Promega, Madison, WI, USA), according to the manufacturer’s protocol for whole-genome sequencing. A long-read library with high-quality genomic DNA of ≥20 μg was prepared using the SMRTbell Express Template Prep Kit 2.0 (Pacific Biosciences, Menlo Park, CA, USA) and sequenced on a PacBio Sequel System using one SMRT cell.

### 2.3. De Novo Genome Assembly and Polishing

PacBio SMRT long reads were assembled using CANU (version 2.2) [20] with the following primary parameters: genomeSize = 45M, minReadLength = 1000, minOverlapLength = 500, rawErrorRate = 0.3, and correctedErrorRate = 0.045. For assembly polishing, Illumina short reads of the two samples (described in ‘Discovery of SNVs in Materials and Methods’) were aligned to the CANU assembly data using BWA-MEM (version 0.7.17) [21], and errors in nucleotides in the CANU assembly data were then corrected using Pilon (version 1.22) [22] with three rounds of polishing. The resulting assembled contigs were finally anchored to nine pseudochromosomes that were mapped to an ultra-density genetic map of *L. edodes* (Lemap 2.0) with a total length of 810.14 cM [23].

### 2.4. Genome Completeness and Coverage Estimation

The completeness of the genome assemblies of Sanmaru and Sanjo was assessed using BUSCO (version 5.2.2) [24], with 758 BUSCO markers from the fungi_odb10 database. For estimation of the sequence coverage, Illumina short-insert paired-end reads were realigned to the draft genome assembly using BWA-MEM (version 0.7.17). Heterozygous nucleotides were identified using SAMtools mpileup (version 1.19) [25].

### 2.5. Genome Annotation

A combination of ab initio and evidence-based approaches were employed for the gene prediction of Sanmaru and Sanjo. The assembled genome was pre-masked for repetitive DNA sequences using RepeatMasker (version 4.0.6) (http://www.repeatmasker.org/, accessed on 22 August 2024). De novo prediction was performed using BRAKER2 (version 2.1.6) [26] with protein evidence. BRAKER2 aligns the protein sequences of *L. edodes* B17 [27] to the two assembled genomes using GenomeThreader (version 1.7.3) [28], trains AUGUSTUS (version 3.3.1) [29] based on splice alignments, and predicts genes with AUGUSTUS using protein homology information. Functional annotation for predicted genes was performed based on homology-based searches with UniProt/SwissProt and NCBI non-redundant (NR) databases using BLASTP (version 2.3.0+) [30], with a cutoff E-value of 1E-06. Protein domains were also identified using InterProScan (version 5.19-58.0) [31]. Genes were searched for families of structurally related catalytic and carbohydrate-binding modules of enzymes that degrade, modify, or create glycosidic bonds using CAZy (http://www.cazy.org, accessed on 22 August 2024) [32]. Additionally, tRNA- and miRNA-like sequences were predicted using tRNAscan-SE (version 1.4 alpha) [33] and Infernal (version 1.1.1) [34]. A sequence comparison of the two genomes was performed using MCScanX (https://github.com/wyp1125/MCScanX, accessed on 22 August 2024) and SyMAP (version 4.2) [35]. The resulting synteny blocks were visualized in a Circos plot. Genome rearrangement events, such as translocations between the two genomes, were verified using MUMmer (version 3.9.4) [36]. All of the protein datasets of the two species were clustered into paralogues and orthologues using OrthoMCL (version 2.0.9) [37], with an inflation parameter of 1.5. Functional annotation for genes showing species-specific expression was performed using DAVID [38], and relevant gene ontology (GO) terms were selected with a cutoff EASE score < 1 × 10^−2^.

### 2.6. Discovery of Single Nucleotide Variants (SNVs)

DNA was extracted from cultured mycelia samples, including Sanmaru 1-33, Sanmaru 1-42, Sanjo 502-19, Sanjo 502-23, M1-6, M1-21, M1-15, and M1-20. Mycelia were frozen in liquid nitrogen and ground into powder. DNA extraction was performed using a Wizard^®^ HMW DNA Extraction Kit (Promega, Singapore) according to the manufacturer’s instructions. The quality of extracted DNA was assessed using the 260/280 absorbance ratio and was found to be within an acceptable range (1.8–2.0). DNA sequence libraries were prepared from 1 µg input DNA using a TruSeq Nano DNA Sample Prep Kit (Illumina, Inc., San Diego, CA, USA), following the manufacturer’s instructions. Sheared DNA fragments were subjected to end-repairing, A-tailing, adaptor ligation, and amplification with clean-up. The libraries were then subjected to paired-end sequencing with a 150 bp read length using the Illumina NovaSeq 6000 platform (Illumina). The quality scores of raw reads were assessed with FastQC (v0.11.9). The reads were processed for quality using Sickle (v1.33, with the following criteria: (i) discard low-quality reads with >10% unknown bases (marked as ‘N’), and (ii) discard reads with >40% low-quality bases (quality score < 20). Clean reads of each sample were aligned to the draft genome of Sanmaru using BWA (v0.7.17), and duplicate reads generated by PCR were removed using Picard Tools (v1.98). SNV calling was performed using the Genome Analysis ToolKit (GATK) (v4.0.5.1) with the following criteria: (i) identification of bi-allelic SNVs, (ii) minimum mapped read depth ≥ 10, (iii) minimum genotype quality ≥ 60, and (iv) no missing alleles among samples. SNPs were annotated using SnpEff (v4.1) with gene annotations. Variants were represented in standard VCF format. The frequency of SNVs was calculated along a 10 kb sliding window, with a step size of 10 kb to identify genomic variant blocks characterized by a high concentration of SNVs between resistance and sensitivity to high temperature.

### 2.7. CAPS Marker Development

The variants used to develop the CAPS marker were selected by prioritizing homozygous missense SNPs present in the coding region (Appendix A). CAPS marker sequences were amplified using 20 ng of genomic DNA as a template. The PCR reaction consisted of an initial denaturation step at 95 °C for 3 min, followed by 35 cycles of amplification (30 s at 95 °C, 30 s at 58 °C, and 20 s at 72 °C), with a final annealing step at 72 °C for 5 min. The amplified target sequences were subjected to Sanger sequencing (Cosmo Genetech, Seoul, Republic of Korea) to validate the mutations. Validated mutations were analyzed using dCAPS Finder 2.0 (http://helix.wustl.edu/dcaps/dcaps.html, accessed on 22 August 2024) to select marker-specific restriction enzymes. PCR-amplified marker sequences of the five cultivars included in the investigation were subjected to restriction enzyme treatment applicable to CAPS. After restriction enzyme digestion, electrophoresis was performed on a 2.5% agarose gel to determine the relative size of each DNA fragment.

## 3. Results

### 3.1. Phenotypic Characterization of High-Temperature Tolerance in Lentinula edodes Cultivars

To genetically define the high-temperature tolerance trait in pyogo cultivars, we first evaluated the phenotype of fruiting body formation at both high and low temperatures. We selected the high-temperature cultivars Sanmaru1 and Sanmaru2, as well as the low-temperature cultivars Sanjo501, Sanjo502, and Kinko135, which are widely cultivated in South Korea. We carefully re-evaluated the temperature range for fruiting body formation for these cultivars to determine the critical temperature for this developmental process. Our analysis confirmed that Sanmaru1 and Sanmaru2 formed fruiting bodies at temperatures ranging from 10 to 29 °C, Sanjo501 from 7 to 18 °C, Sanjo502 from 6 to 20 °C, and Kinko135 from 7 to 15 °C (Figure 1A). At a relatively low temperature of 10 °C, all five cultivars were able to form fruiting bodies. However, at high temperatures above 25 °C, only Sanmaru1 and Sanmaru2 were capable of forming fruiting bodies (Figure 1B). Therefore, using 25 °C as the high-temperature reference, we classified Sanmaru1 and Sanmaru2 as high-temperature-tolerant cultivars, while Sanjo501, Sanjo502, and Kinko135 were classified as high-temperature-sensitive cultivars.

### 3.2. Genetic and Genomic Basis of High-Temperature Tolerance

To identify the high-temperature tolerance traits underlying the genetic basis of fruiting body formation, we constructed inbred and hybrid populations centered on Sanmaru1 (mating type: A1 and A11) and Sanjo502 (mating type: A1 and A5) to establish phenotypic and genetic mapping populations (Appendix A). Under high-temperature conditions, all self-mating dikaryotic strains (inbred lines) of Sanmaru1 formed fruiting bodies (Appendix A). However, those of Sanjo502 could not form fruiting bodies (Appendix A). To further investigate the genetic characteristics associated with high-temperature tolerance for fruiting body formation, we constructed hybrid F1 populations of Sanmaru1 and Sanjo502 (Appendix A). We then generated F2 populations by self-crossing hybrid dikaryotic M1 strains of Sanmaru1-33 and Sanjo502-23 monokaryotic strains (Appendix A). Through this genetic approach, we obtained several high-temperature-tolerant (M1-6 and M1-21) and -sensitive (M1-15 and M1-20) monokaryotic F2 strains (Appendix A).

Subsequently, we performed a genome assembly to identify the high-temperature resistance genes in Sanmaru1 and conducted a comparative genomics analysis. A total of 11.5 Gb (an average sequencing coverage of 248.9×) and 11.9 Gb (258.7×) of PacBio long reads from Sanmaru1-33 and Sanjo502-19 were generated, respectively (Appendix A). The genome assemblies of Sanmaru and Sanjo resulted in 47.1 Mb and 46.5 Mb draft genomes with 12 contigs (contig N50 length of 4.6 Mb) and 11 contigs (contig N50 length of 5.4 Mb), respectively (Figure 2A; Appendix A). Additionally, the mitochondrial genomes of the two species were also assembled into single circular molecules of 121,489 bp and 121,857 bp (Appendix A). This represents a significant improvement over previous assemblies such as the work by Chen et al. [11], which resulted in over 340 scaffolds, and the study by Shim et al. [12] that achieved 31 scaffolds. Our use of PacBio sequencing enabled us to construct a chromosome-level genome assembly, providing greater accuracy in identifying genomic variations and improving the resolution for functional and evolutionary studies. In this study, we aligned the genome assemblies of Sanmaru 1-33 and Sanjo 502-23 to the pseudochromosome data (Lemap2.0) that were anchored to genetic linkage groups, as reported by Zhang et al. (2021) [23]. This alignment was to assign chromosome numbers of *L. edodes* to the contigs of the two genomes based on Lemap2.0. The contigs of the Sanmaru 1-33 and Sanjo 502-23 genomes were aligned to the 66 scaffolds of Lemap2.0, which were assembled using short reads, through MCScanX and SyMap. MCScanX was employed to identify putative homologous genomic regions centered around protein-coding genes, and SyMap was used to further verify synteny blocks at the DNA sequence level between Lemap2.0 and the two genomes, ultimately determining chromosome number assignments. From this alignment, we observed that the contigs of the two genomes covered 94.3% of the assembled sequences in Lemap2.0. The observed discrepancy (5.7%) likely arises from the differences in assembly methods—our study utilized long-read de novo assembly with high sequencing depth, whereas Lemap2.0 was assembled using short-read de novo methods, potentially leading to misassembles that could have limited genome-to-genome comparisons. Alternatively, this discrepancy may also reflect the genetic diversity between the L. edodes L54A strain (used in Lemap2.0) and the Sanmaru or Sanjo strains. Based on this alignment, we constructed chromosome-level pseudochromosome maps for Sanmaru 1-33 and Sanjo 502-23.

The quality of the genome assembly was assessed using Benchmarking Universal Single-Copy Orthologs (BUSCO). The draft genome assemblies of Sanmaru and Sanjo captured 94.9% and 94.6% of the complete BUSCOs with the fungi_odb10 database, respectively, indicating the high completeness of our assembly data in comparison with Lemap 2.0 (left panel in Figure 2B). Notably, the proportion of complete and single-copy BUSCOs accounted for about 93%. Considering the number of assemblies, those of Sanmaru and Sanjo were superior to Lemap 2.0 (right panel in Figure 2B). The whole-genome comparison revealed four genomic rearrangements between Sanmaru and Sanjo (“synteny block” layer in Figure 2A; Appendix A). The rearrangements were as follows: alignment between positions 72.6 and 1310 kb on Chr2_1 of Sanmaru and positions 5389.2 and 6802.9 kb on Chr_7 of Sanjo; alignment between positions 18.8 and 1169.9 kb on Chr2_2 of Sanmaru and positions 12.4 and 1287.0 kb on Chr_1 of Sanjo; alignment between positions 9.5 and 1101.3 kb on Chr1_1 of Sanmaru and positions 137.8 and 999.8 kb on Chr_3 of Sanjo; and alignment between positions 1171.7 and 2399.3 kb on Chr1_1 of Sanmaru and positions 630.8 and 1832.3 kb on Chr2_2 of Sanjo. This genomic rearrangement was also validated using JCVI [39], a tool designed for visualizing synteny at the chromosome level when comparing different related genomes (Appendix A). This result indicates genomic changes since the divergence from a common ancestor of *L. edodes*.

### 3.3. Genomic Characterization and Functional Analysis

Employing ab initio and evidence-driven gene prediction methodologies, we predicted a non-redundant set of 13,726 protein-coding genes in the Sanmaru genome, with an average gene length of 2.1 kb, an average exon length of 1.6 kb, and an average of 5.6 exons per gene (Table 1). In the Sanjo genome, 14,247 non-redundant protein-coding genes were predicted with similar characteristics (Table 1). The genic regions accounted for 69.5% and 68.4% of the draft genomes of Sanmaru and Sanjo, respectively. About 97% of predicted genes in both genomes showed homology with gene models in the UniProt (55%), NCBI non-redundant (NR) (95%), and InterPro (88%) databases (Table 1). To investigate species-specific and shared genes between Sanmaru and Sanjo, we analyzed orthologous gene clusters. This analysis revealed that 9561 out of 11,057 orthologous gene clusters were shared between the two species, while 568 and 928 orthologous gene clusters were species-specific to Sanmaru and Sanjo, respectively (Figure 2C). This highlighted an enrichment of genes related to DNA recombination functions (*p*-value: 2.1 × 10^−7^) in Sanmaru (Figure 2C). Unlike Sanmaru, genes related to the lysine biosynthetic process via aminoadipic acid (*p*-value: 6.1 × 10^−3^) were significantly enriched in Sanjo (Figure 2C), suggesting the evolutionary conservation of the AAA (alpha-aminoadipate) pathway of lysine biosynthesis in general Agaricomycetes, including wild or cultivated species [40,41].

We also searched for carbohydrate-active enzymes (CAZymes) to identify genes encoding enzymes involved in macromolecular carbohydrates specific to Sanmaru or Sanjo. We annotated the CAZymes in the genomes of Sanmaru 1-33, Sanjo 502-19, and B17 using dbCAN2 [42]. Among the total genes, 647 (3.9%) in B17, 656 (4.13%) in Sanmaru 1-33, and 685 (4.18%) in Sanjo 502-19 were annotated as CAZymes. The proportion of CAZyme genes relative to the total number of genes is very similar across the strains. Additionally, the number of genes in each CAZyme class is highly comparable (Appendix A). This suggests that gene prediction for each strain was performed adequately, allowing for a confident comparison of CAZyme class functions between Sanmaru 1-33 and Sanjo 502-19. Based on these results, we identified CAZyme-annotated genes within strain-specific gene clusters based on orthologous gene cluster analysis for Sanmaru 1-33 and Sanjo 502-19 (Appendix A).

In this analysis, glycoside hydrolase (GH) families containing potential β-glucosidases were more abundant in Sanjo than in Sanmaru (Figure 2D). A previous study by Li et al. [43] suggested that GH families, which were analyzed from 21,244 bacterial, 424 archaeal, 456 viral, and 352 eukaryotic genomes in the CAZy database, have relatively narrow environmental distributions, with the highest abundance typically found in host-associated environments and a preference for moderate low-temperature and acidic environments. Moreover, they reported that GH subfamilies with a (β/α)8-barrel structure exhibit high resistance to high temperatures and are highly robust to mutations [43]. Remarkably, GHs with the (β/α)_8_-barrel structure, including *AMY1*, *GUN4*, and *RHG4*, were identified in our study, suggesting that these enzymes may enhance the functional diversity of Sanmaru at high temperatures. Additionally, the robustness to mutations observed in these GH subfamilies may be linked to the occurrence of a 560 kb variant block (VB) between Sanmaru and Sanjo. In addition to GHs, the relative abundance of glycosyltransferase (GT), carbohydrate esterase (CE), and auxiliary activity (AA) families were identified in Sanjo. This result may suggest functional differences between Sanmaru and Sanjo in utilizing primary carbon sources, including lentinan biosynthesis, for *L. edodes* growth and development. Additionally, a total of 10.3 Mb and 9.6 Mb of transposable elements (TEs) were identified in Sanmaru and Sanjo, respectively, accounting for 21.8% and 20.6% of their genomes (Appendix A). Among the class I elements, long terminal repeat (LTR) retrotransposons and LINEs for non-LTR retrotransposons were abundant in both genomes.

### 3.4. Genetic Variation and Structural Analysis Associated with High-Temperature Tolerance

To investigate genetic variation between resistance and sensitivity to high temperature, samples derived from Sanmaru (resistant; including Sanmaru1-33, Sanmaru1-42, M1-6, and M1-21) and Sanjo (sensitive; including Sanjo502-19, Sanjo502-23, M1-15, and M1-20) were re-sequenced (Appendix A). Based on the draft genome of Sanmaru, we analyzed single nucleotide variants (SNVs) between the two groups. We focused on regions where SNVs occur in a structurally consecutive manner (regions with variations spanning over 50 kb) and where homozygous variants oppose each other for the two traits. From the genome-wide SNP data, we extracted genotypes where the resistant samples had homozygous reference (REF) alleles, and the sensitive samples had homozygous alternative (ALT) alleles. As a result, we identified a large variant block (VB) approximately 560 kb in length, with densely distributed SNVs on chromosome 9 between the Sanmaru and Sanjo groups (Figure 3A). Within the VB region, a high frequency of SNVs in genic regions was observed: 1.7 for missense, 3.1 for synonymous, and 8.7 for intron variants per gene (Figure 3B). This variant block is likely associated with the thermo-tolerant trait for fruiting body formation in Sanmaru compared to Sanjo. This genetic event suggests the hypothesis that genetic recombination does not occur to preserve high-temperature resistance in Sanmaru. In connection with the functional annotation of all genes within the VB region, we identified an enrichment of genes associated with DNA repair (*p*-value = 0.006), including *PSO2*, *CND2*, *DPOD*, *PCNA*, *ARP9*, and *MMS22* (Figure 3C). Additionally, structural differences at the 5′-upstream region of the phospholipase C 1-encoding gene (*PLC1*) in the VB region were identified (Figure 3D).

### 3.5. Development and Validation of CAPS Markers for High-Temperature Tolerance

We designed CAPS markers for the proper selection of the high-temperature tolerance-associated VB region. Seven SNPs were selected as potential CAPS candidates located on GENE03306, slc16a10, GENE03321, GENE03324, GENE03326, MNS3, and GENE00389, altering recognition sites for HpaII, ScrFI, AluI, BglII, and NlaIII restriction enzymes (Figure 4A and Appendix A). For amplification of the marker sequences, primer sets RL-LE-306 (234 bp), RL-LE-307 (304 bp), RL-LE-310 (326 bp), RL-LE-311 (330 bp), RL-LE-312 (329 bp), RL-LE-314 (395 bp), and RL-LE-316 (228 bp) were designed (Table 2). The efficacy of CAPS markers was validated for both high-temperature-tolerant (Sanmaru1 and Sanmaru2) and -sensitive (Sanjo501, Sanjo502, and Kinko135) cultivars. As presented in Figure 4B, all seven CAPS markers successfully discriminated the high-temperature-tolerant or -sensitive SNPs in the VB regions of the selected cultivars (Figure 1 and Figure 4B).

## 4. Discussion

The discovery and genetic analysis of a genome-wide variant block linked to heat tolerance traits in *Lentinula edodes* provide essential insights into the genetic improvement and breeding processes of pyogo mushrooms. Our study focused on phenotypic screening under controlled high-temperature conditions and generated a mapping population of *L. edodes* (Figure 1 and Appendix A). By comparing the genomes of individuals exhibiting high-temperature tolerance with those showing high-temperature sensitivity, we successfully identified a specific genomic variant block region on chromosome 9 that is strongly associated with a thermotolerance phenotype (Figure 2 and Figure 3). The variant block on chromosome 9, spanning approximately 560 kilobases, is of particular interest due to its conservation among high-temperature tolerant cultivars or monokaryotic strains (Figure 3A). The genes within this variant block likely play crucial roles in maintaining cellular functions and protecting against heat-induced damage through DNA repair pathways (Figure 3C). This result indicates that DNA damage tolerance due to heat stress may involve recombination-mediated mechanisms to prevent fork breakage, a leading cause of genome instability [40]. The presence of potential genes implicated in heat stress response and thermoregulation within the variation block further underscores its significance in thermotolerance. These findings suggest that the potential genetic factors associated with high-temperature tolerance found in Sanmaru1 and Sanjo502 are likely homozygous. Understanding the specific molecular pathways and functions of the candidate genes within this variant block will provide deeper insights into the mechanisms underlying thermotolerance in *L. edodes*. These findings open new avenues for breeding programs aimed at improving thermotolerance in pyogo mushrooms. Marker-assisted selection can be employed to identify individuals carrying the variant block associated with thermotolerance, leading to the development of robust and resilient strains that can withstand high-temperature environments.

This study demonstrates the successful application of comparative whole-genome assembly, SNP discovery, marker development, and CAPS analysis to discriminate thermotolerance-related phenotypes of pyogo cultivars. Our results also indicate that the VB identified in this study is associated with high-temperature tolerance for fruiting in *L. edodes* and that the developed CAPS markers can be used to distinguish proper temperature types. Moreover, the marker development pipeline presented in this study has the potential to be effectively utilized for the design of markers that can discriminate among various cultivars of other edible mushrooms. The genetic variants associated with high-temperature tolerance of fruiting bodies identified in this work offer valuable insights for enhancing our understanding of the temperature preferences of *L. edodes*. Linkage disequilibrium analysis and the generation of haplotype blocks would be crucial for quantitative trait studies. The putative SNPs linked to the ability of fruiting bodies to tolerate high temperatures are valuable for investigating the temperature preference of *L. edodes*.

The methodology employed in this research to uncover potential polymorphisms related to traits through the comparison of the genome sequences of individuals with contrasting phenotypes has the potential to be a valuable genetic tool for identifying other desirable characteristics in fungal genetic analysis. Due to its longer cultivation time compared to other mushrooms, the process of developing new pyogo cultivars through mating and evaluating the morphological phenotype of fruiting bodies is time-consuming [44]. Cultivar-specific markers serve to ensure the distinct characteristics of cultivars, prevent illegal utilization of strains, and mitigate the risk of crossbreed contamination. Markers related to specific traits can be efficiently utilized for genetic breeding processes [45]. Markers positioned within the coding region of a gene can be exploited more effectively due to their direct impact on gene function and associated pathways [46].

Molecular markers have been demonstrated to be effective in genetic diversity and evolutionary analyses, identifying candidate genes that regulate different mechanisms and traits, and in marker-assisted breeding [47]. The CAPS markers developed in this study were positioned within the intragenic regions of the high-temperature-associated VB. The potential genes linked with high-temperature tolerance are PLC1 (Figure 3D) and rho guanine nucleotide exchange factor (RhoGEF) [48]. In particular, the fungal Ras family genes play roles in various cellular processes such as nutrition response, stress response, cell cycle regulation, and cyclic adenosine monophosphate synthesis. Further study of the functional roles of RhoGEF and PLC1 is likely to enhance our understanding of high-temperature tolerance during the reproductive phase transition of *L. edodes*.

## Figures and Tables

**Figure 1 jof-10-00628-f001:**
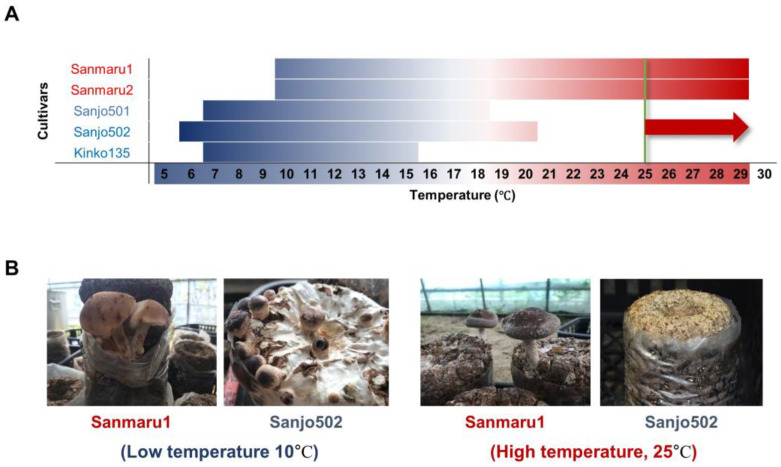
Strain selection for analysis of high-temperature tolerance of *Lentinula edodes*. (**A**) Fruiting temperature range of high-temperature-tolerant (Red lines, Sanmaru1 and Sanmaru2) and high-temperature-sensitive (Blue lines, Sanjo501, Sanjo502, and Kinko135) cultivars. Red arrow indicates temperature range for thermo-tolerant cultivars, including Sanmaru1 and Sanmaru2. (**B**) Fruiting phenotypes of Sanmaru1 and Sanjo502 grown under low (10 °C) and high (25 °C) temperature conditions. Sanmaru1 showed fruiting body formation under both high and low temperature conditions, whereas Sanjo502 showed fruiting body formation under only low temperature conditions.

**Figure 2 jof-10-00628-f002:**
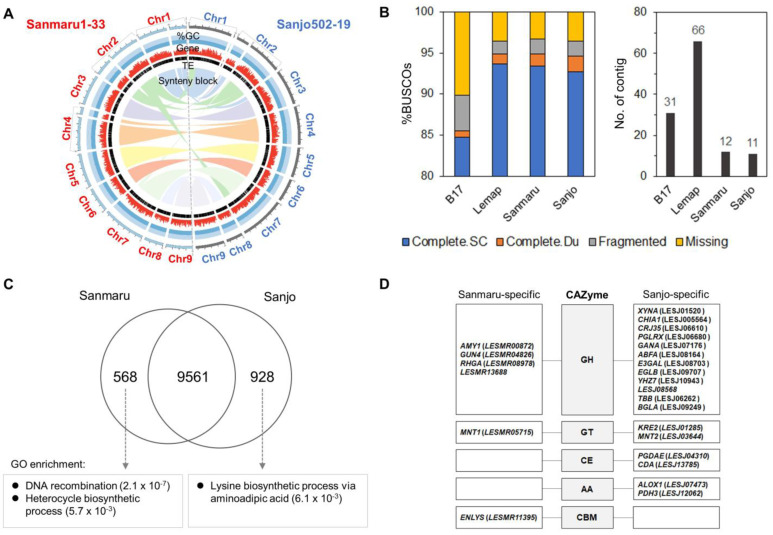
Overview of whole-genome sequences of monokaryotic Sanmaru1-33 and Sanjo502-23. (**A**) Whole-genome sequence assembly of Sanmaru1-33 (Sanmaru) and Sanjo502-23 (Sanjo) strains. A total of 12 and 11 contigs of Sanmaru and Sanjo, respectively, were anchored to the chromosome-level genome map (Lemap 2.0) reported by Zhang et al. [23]. Each track in the Circos plot, from the outside in, represents the following: each chromosome anchored to Lemap 2.0 (1st track); GC content (%) (2nd track); gene density (3rd track); transposable element (TE) density (4th track); and synteny blocks conserved between Sanmaru and Sanjo (5th track). (**B**) Quality assessment of Sanmaru and Sanjo assemblies. The left panel shows BUSCO scores for B17 [12], Lemap, Sanmaru, and Sanjo assemblies. The results are categorized as complete and single-copy (Complete.SC), complete and duplicated (Complete.Du), fragmented, or missing BUSCOs. The right panel shows the number of assembled contigs in these four species. (**C**) Overlap of gene orthologous clusters between Sanmaru and Sanjo. Significant GO-enriched terms for species-specific orthologous genes with *p*-values are presented in the bottom rectangular box. (**D**) Carbohydrate-active enzyme (CAZyme) annotation for genes specific to Sanmaru and Sanjo.

**Figure 3 jof-10-00628-f003:**
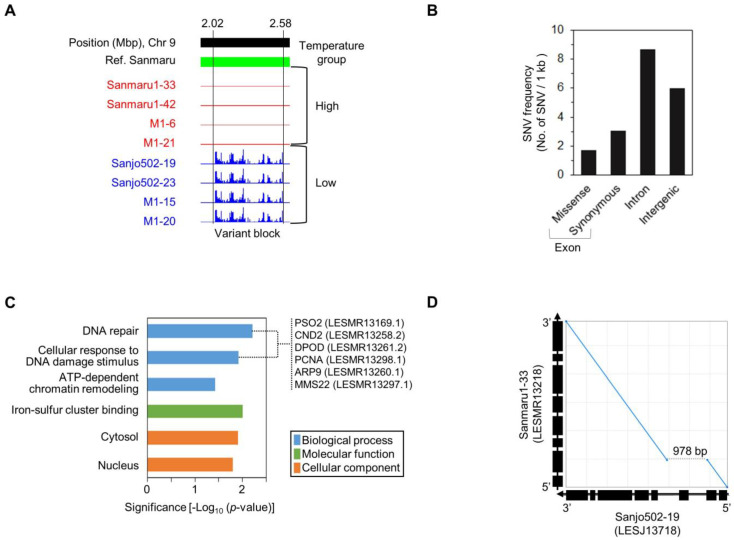
Genomic differences between Sanmaru and Sanjo. (**A**) Identification of a large variant block (VB) that predominantly consists of SNVs on chromosome 9 between Sanmaru and Sanjo relatives. (**B**) Distribution of SNVs identified in the VB region. The frequency of SNVs was calculated by dividing 1 kb of genomic features (i.e., exons, introns, or intergenic regions) in the 560 kb VB region by the number of SNVs. SNVs were categorized as missense, synonymous, intron, and intergenic. (**C**) Functional annotation of genes within the VB region. (**D**) Gene structural variation of the phospholipase C 1-encoding gene (PLC1) in the VB region. In the dot plot (978 bp) comparing Sanmaru and Sanjo, structural differences in PLC1 were identified.

**Figure 4 jof-10-00628-f004:**
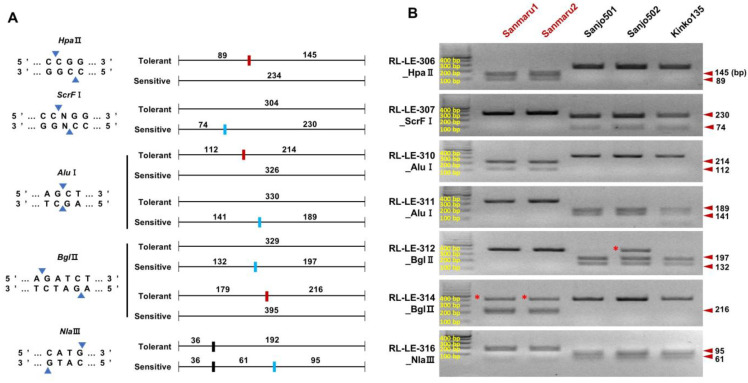
CAPS markers associated with high-temperature tolerance of fruiting body formation. (**A**) The restriction enzymes used in the development of CAPS markers and schematic diagrams illustrating the position of the restriction enzyme recognition sites in the marker sequence of each allele. The fragment sizes of PCR and CAPS products are shown. Additional restriction enzyme recognition sites unrelated to the target SNP (RL-LE-316_NlaIII) are indicated by black boxes. The red and blue boxes represent the restriction enzyme-recognizing and non-recognizing sites present in the high-temperature tolerance allele, respectively. (**B**) The cleaved fragment patterns of each CAPS marker among five *L. edodes* cultivars. Sanmaru1 and Sanmaru2 show the high-temperature-tolerant allele-specific fragment pattern. Arrowheads indicate the position of putative high-temperature-tolerant allele-specific fragments. Asterisks represent undigested bands.

**Table 1 jof-10-00628-t001:** Gene prediction and annotation in the genomes of Sanmaru and Sanjo.

	Quantification	
	Sanmaru 1-33	Sanjo 502-19
Total no. of gene models predicted	15,900	16,380
Unique gene models (No.)	13,726	14,247
Genes with isoforms (No.)	2174	2133
Average gene length (bp)	2059 bp	1942 bp
Total bases of gene models (Mbp)	32.74 Mbp	31.81 Mbp
%Genes in the draft genome	69.49	68.41
No. of exon	88,779	91,869
Average no. of exon per gene	5.58	5.60
Average exon length (bp)	285 bp	274 bp
%Exon fraction in the draft genome	53.75	54.28
No. of intron	72,879	75,489
Average no. of intron per gene	4.58	4.60
Average intron length (bp)	101 bp	86 bp
%Intron fraction in the draft genome	15.74	14.13
Functional annotated genes	13,331 (97.12%)	13,851 (97.22%)
No. hit to UniProt	7554 (55.03%)	7718 (54.11%)
No. hit to NCBI NR	13,032 (94.94%)	13,541 (95.04%)
No. hit to InterPro	12,120 (88.29%)	12,487 (87.64%)

**Table 2 jof-10-00628-t002:** Characteristics of CAPS markers used for variant detection.

Primer Name	Sequence (5′-3′)	Product Size (bp)	Restriction Enzyme	Digested Fragment Size (bp)	
				High-Temperature Sensitive	High-Temperature Tolerant
RL-LE-306 F	GGTAACAGTCGTCAGGTGAAAG	234	HpaII (+)	234	89/145
RL-LE-306 R	CTCCGGGATCAAAGGTTACTAC				
RL-LE-307 F	GAACCTAGCTAGACTGACGCAT	304	ScrFI (−)	74/230	304
RL-LE-307 R	GATAGATAGTCATACGGCCCTC				
RL-LE-310 F	ATTCGACTCTCTCTCTTCCTCC	326	AluI (+)	326	112/214
RL-LE-310 R	CTTCTTGTCCACTCTGAGTTCC				
RL-LE-311 F	GACTAGCAACAGGTTACCTCTCC	330	AluI (−)	141/189	330
RL-LE-311 R	CCTATACCACGAGAGAAGAGTAGG				
RL-LE-312 F	GACAGCCTACATGGTGATGAG	329	BglII (−)	132/197	329
RL-LE-312 R	CTAGTGTCAAGAATGCACTCCC				
RL-LE-314 F	GGGATACTACTTCTCTCCAACG	395	BglII (+)	395	179/216
RL-LE-314 R	CTACCGGACGACTGAACTTCT				
RL-LE-316 F	CTACACGCCTAGCTCAGAAGTT	192	NlaIII (−)	36/61/95	36/156
RL-LE-316 R	CTAACAGCTCTAGGATAGGCAGAC				

Restriction enzyme recognition site abolished (−) or created (+) due to mutations in the marker sequence was represented.

## Data Availability

The draft genome assembly data utilized in this investigation have been submitted to the NCBI SRA database and are accessible to the public under BioProject accession PRJNA937439. This includes the Sanjo genome assembly (GCA_030770125.1) and the Sanmaru genome assembly (GCA_030770075.1). Illumina DNA paired-end sequencing data can be accessed via BioProject accession PRJNA938156, which contains data from eight samples: SRR23604074, SRR23604075, SRR23604076, SRR23604077, SRR23604078, SRR23604079, SRR23604080, and SRR23604081.

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
