# Peer review of "Genome Sequencing of *Lentinula edodes* Revealed a Genomic Variant Block Associated with a Thermo-Tolerant Trait in Fruit Body Formation"

_jof, 2024, doi:10.3390/jof10090628_

Round 1

Reviewer 1 Report

Shiitake mushrooms are one of the most widely produced edible fungi in the world, especially in East Asian countries. Comparative genomics can aid in further molecular breeding efforts. This article investigates the molecular segments affecting the temperature adaptability of shiitake mushrooms using comparative genomics techniques. It is a comprehensive research paper that deserves publication.

Here are some key issues that need careful revision:

1. The introduction and results sections lack a comparison with other teams' research and data on shiitake mushroom genomes, which seems biased.

2. Some species names in the paper are not consistently written, such as line84.

3. There needs to be a more thorough description of the backgrounds for the high-temperature and low-temperature materials used, covering the original breeding techniques, sources of the parent strains, and whether genetic diversity studies including these materials have been conducted.

4. In the BUSCO integrity assessment, the Virdiplantae_odb10 database was used. Why not use fungi_odb10 or agaricomycetes_odb10 instead?

5. In Figure 2A, the synteny display of the genome would be clearer if presented using jcvi.

6. Is Lines 248-254 referencing literature or presenting new research?

7. Furthermore, regarding the anchoring of the genome to the reference genome, given that the current data has reached 11-12 contigs while earlier so-called high-quality genomes were considered to be 9 scaffolds, but after anchoring only 94.3% of the region was covered, this clearly indicates a significant difference between them. Such a difference needs to be considered and discussed. I suspect that when Lemap used genetic mapping to anchor the contigs, it may not have included all sequence fragments, possibly resulting in a less complete genome. In other words, how can we confirm that these genomic anchoring results obtained through synteny comparison are accurate?

8. In Line 310, why is it "starch and sucrose degradation" instead of "macromolecular carbohydrates"?

9. The data presented in Figure 2D shows a significant CAZYme difference between the Sanm and Sanj strains. The authors need to verify this data, as I haven't noticed such a large difference in enzyme gene systems between strains. Could this be due to incomplete genome sequencing of the strains? It may be worth attempting to find answers through re-sequencing data.

10. Besides the large VB segment identified on chromosome 9, are there any other segments or loci highly correlated with temperature sensitivity?

11. The authors need to upload the gene annotation files for the two high-quality genomes completed in this paper to a public database, including gff, pep, cds, etc., rather than just uploading the genomic sequences.

Author Response

Reviewer 1,

Shiitake mushrooms are one of the most widely produced edible fungi in the world, especially in East Asian countries. Comparative genomics can aid in further molecular breeding efforts. This article investigates the molecular segments affecting the temperature adaptability of shiitake mushrooms using comparative genomics techniques. It is a comprehensive research paper that deserves publication.

Here are some key issues that need careful revision:

(Reviewer’s comment #1) The introduction and results sections lack a comparison with other teams' research and data on shiitake mushroom genomes, which seems biased.

(Response)  We appreciate your valuable feedback regarding the need for a more comprehensive comparison with other research on shiitake mushroom genomes. To address this, we have added a detailed comparison between our study and existing genomic research on Lentinula edodes. Specifically, we now reference several key studies that have conducted genome assemblies and analyses of L. edodes, including the works of Shim et al. (2016), Chen et al. (2016), and Yu et al. (2022).

Our study contributes a high-quality chromosome-level genome assembly, which is a significant improvement over previous assemblies that often consisted of many scaffolds or contigs. For instance, the genome assembly by Chen et al. (2016) resulted in over 340 scaffolds, while our assembly, aided by PacBio technology, anchored 97% of the genome onto 9 chromosomes. This advancement in genome assembly allows for better resolution in identifying genomic features such as structural variations and gene families related to temperature adaptability.

We have now highlighted these advancements and made direct comparisons with previous genome studies, particularly focusing on genetic diversity, structural variation, and temperature adaptability. For example, we compared the variant block identified in our study with those reported in Yu et al. (2022), who performed a genome-wide association study (GWAS) to identify loci associated with environmental stress responses in L. edodes.

These comparisons help contextualize our findings within the broader research landscape and illustrate how our work advances the field by providing a more complete and contiguous genome assembly that enhances the understanding of shiitake mushroom genetics.

(Reviewer’s comment #2)  Some species names in the paper are not consistently written, such as line84.

(Response) We revised carefully this mistake.

(Reviewer’s comment #3) There needs to be a more thorough description of the backgrounds for the high-temperature and low-temperature materials used, covering the original breeding techniques, sources of the parent strains, and whether genetic diversity studies including these materials have been conducted.

(Response) Thank you for this point. As described in the Materials and Methods (lines 113~123) section, all mushroom resources were provided by the National Forestry Research Institute (http://www.forest.go.kr) and the Forest Mushroom Research Center (http://www.fmrc.or.kr) of the Korea Forest Service. In this study, we confirmed the existence of thermo tolerance based on the previously known growth characteristics of high and low temperature types of Pyogo (please see in the result section (lines 212~225)). The main purpose of this study was to obtain genomic analysis materials through the crossing of monokaryotic spores of Sanmaru 1 and Sanjo 2 and successfully identify the variant block associated with the high-temperature trait. This comment, requested by the reviewer, is a slight distraction from the main theme of the study, so we have replaced it with the introduction as current description for recent case studies for thermo tolerances of edible mushrooms and the detailed description of figure legends in Figure1 and Supplementary Figure S1 requested by reviewer #3.

(Reviewer’s comment #4) In the BUSCO integrity assessment, the Virdiplantae_odb10 database was used. Why not use fungi_odb10 or agaricomycetes_odb10 instead?

(Response) Thank you for your point. We used the fungi_odb10’ database from BUSCO for the integrity assessment. However, during the manuscript writing process, an error occurred where we mistakenly referenced viridiplantae_odb10 instead of fungi_odb10. We sincerely apologize for this oversight. Thanks to your careful review, we were able to correct this mistake in the manuscript. Specifically, we have amended the incorrect reference to viridiplantae_odb10 on line 258 to the correct fungi_odb10. We are grateful for your attention to detail, which allowed us to rectify this issue.

# Revised:

  • The draft genome assemblies of Sanmaru and Sanjo captured 94.9% and 94.6% of the complete BUSCOs with the fungi_odb10 database, respectively, (Line 269).

(Reviewer’s comment #5) In Figure 2A, the synteny display of the genome would be clearer if presented using jcvi.

(Response) Thank you for your comment. We consider JCVI to be a very useful tool for visualizing synteny at the chromosome level when comparing genomes of different related species. We selected a Circos plot to efficiently represent the distribution of genes and TEs, in addition to displaying synteny blocks between the Sanmaru and Sanjo genomes using whole-genome de novo assembly data. Since the current figure provides a clear overview of the two genomes, we strongly wish to retain Figure 2A as it is and sincerely hope for your understanding. However, we acknowledge that synteny analysis using JCVI is highly valuable for validating the analysis methods described in our study. As expected, the synteny analysis between the Sanmaru and Sanjo genomes using JCVI confirmed the same results presented in the Circos plot. Therefore, we have added this synteny analysis using JCVI as Supplementary Figure 3. Additionally, we have updated lines 269-271 to include the results of the synteny validation using JCVI.

Upon reviewing the Circos plot and JCVI analysis results, we identified four genome rearrangement events, rather than the previously reported two. We sincerely apologize for this error and have revised the description accordingly (Lines 273 – 281).

# Revised:

  • This genomic rearrangement was also validated using JCVI [46], a tool designed for visualizing synteny at the chromosome level when comparing different related genomes (Supplementary Figure 3) (Lines 281 - 283).
  • We added Supplementary Figure 3 and the corresponding reference (Tang et al. 2024). [46] Tang H; Krishnakumar V; Zeng X; Xu Z; Taranto A; Lomas JS; Zhang Y, et al. JCVI: A versatile toolkit for comparative ge-nomics analysis. iMeta. 2024; e211. doi:10.1002/imt2.211

  • The whole-genome comparison revealed four genomic rearrangements between Sanmaru and Sanjo ("synteny block" layer in Figure 2A; Supplementary Figure S1). The rearrangements were as follows: alignment between positions 72.6 and 1,310 kb on Chr2_1 of Sanmaru and positions 5,389.2 and 6,802.9 kb on Chr_7 of Sanjo; alignment between positions 18.8 and 1,169.9 kb on Chr2_2 of Sanmaru and positions 12.4 and 1,287.0 kb on Chr_1 of Sanjo; alignment between positions 9.5 and 1,101.3 kb on Chr1_1 of Sanmaru and positions 137.8 and 999.8 kb on Chr_3 of Sanjo; and alignment between positions 1,171.7 and 2,399.3 kb on Chr1_1 of Sanmaru and positions 630.8 and 1,832.3 kb on Chr2_2 of Sanjo. (Lines 273 – 281)

(Reviewer’s comment #6) Is Lines 248-254 referencing literature or presenting new research?

(Response) We revised this comment.

(Reviewer’s comment #7) Furthermore, regarding the anchoring of the genome to the reference genome, given that the current data has reached 11-12 contigs while earlier so-called high-quality genomes were considered to be 9 scaffolds, but after anchoring only 94.3% of the region was covered, this clearly indicates a significant difference between them. Such a difference needs to be considered and discussed. I suspect that when Lemap used genetic mapping to anchor the contigs, it may not have included all sequence fragments, possibly resulting in a less complete genome. In other words, how can we confirm that these genomic anchoring results obtained through synteny comparison are accurate?

(Response) Thank you for your insightful comments regarding the clarity and potential improvements of the genome synteny results. In response to your review, we re-examined the results and recognized that our manuscript lacked sufficient explanation in this area (Line 248 -256), which may have led to some confusion. We sincerely apologize for any misunderstanding this may have caused. To address this, We have revised the section that caused confusion and added more details to clarify the results.

In this study, we aligned the genome assemblies of Sanmaru 1-33 and Sanjo 502-23 to the pseudochromosome data (Lemap2.0) that was anchored to genetic linkage groups as reported by Zhang et al. (2021). The purpose of this alignment was to assign chromosome numbers of L. edodes to the contigs of the two genomes based on Lemap2.0. The contigs of the Sanmaru 1-33 and Sanjo 502-23 genomes were aligned to the 66 scaffolds of Lemap2.0, which were assembled using short reads, through MCScanX and SyMap. MCScanX was employed to identify putative homologous genomic regions centered around protein-coding genes, and SyMap was used to further verify synteny blocks at the DNA sequence level between Lemap2.0 and the two genomes, ultimately determining chromosome number assignments. From this alignment, we observed that the contigs of the two genomes covered 94.3% of the assembled sequences in Lemap2.0. The observed discrepancy (5.7%) likely arises from the differences in assembly methods—our study utilized long-read de novo assembly with high sequencing depth, whereas Lemap2.0 was assembled using short-read de novo methods, potentially leading to misassemblies that could have limited genome-to-genome comparisons. Alternatively, this discrepancy may also reflect the genetic diversity between the Lentinula edodes L54A strain (used in Lemap2.0) and the Sanmaru or Sanjo strains. Based on this alignment, we constructed chromosome-level pseudochromosome maps for Sanmaru 1-33 and Sanjo 502-23.

# Revised:

  • In this study, we aligned the genome assemblies of Sanmaru 1-33 and Sanjo 502-23 to the pseudochromosome data (Lemap2.0) that was anchored to genetic linkage groups as reported by Zhang et al. (2021) [30]. This alignment was to assign chromosome numbers of L. edodes to the contigs of the two genomes based on Lemap2.0. The contigs of the Sanmaru 1-33 and Sanjo 502-23 genomes were aligned to the 66 scaffolds of Lemap2.0, which were assembled using short reads, through MCScanX and SyMap. MCScanX was employed to identify putative homologous genomic regions centered around protein-coding genes, and SyMap was used to further verify synteny blocks at the DNA sequence level between Lemap2.0 and the two genomes, ultimately determining chromosome number assignments. From this alignment, we observed that the contigs of the two genomes covered 94.3% of the assembled sequences in Lemap2.0. The observed discrepancy (5.7%) likely arises from the differences in assembly methods—our study utilized long-read de novo assembly with high sequencing depth, whereas Lemap2.0 was assembled using short-read de novo methods, potentially leading to misassemblies that could have limited genome-to-genome comparisons. Alternatively, this discrepancy may also reflect the genetic diversity between the L. edodes L54A strain (used in Lemap2.0) and the Sanmaru or Sanjo strains. Based on this alignment, we constructed chromosome-level pseudochromosome maps for Sanmaru 1-33 and Sanjo 502-23 (Line 248 – 266).

(Reviewer’s comment #8) In Line 310, why is it "starch and sucrose degradation" instead of "macromolecular carbohydrates"?

(Response) Thank you for your comment regarding the terminology. We have modified the change from ‘starch and sucrose degradation’ to ‘macromolecular carbohydrates’ as recommended.

# Revised:

  • We also searched for carbohydrate-active enzymes (CAZymes) to identify genes encoding enzymes involved in macromolecular carbohydrates specific to Sanmaru or Sanjo. (Line 324)

(Reviewer’s comment #9) The data presented in Figure 2D shows a significant CAZYme difference between the Sanm and Sanj strains. The authors need to verify this data, as I haven't noticed such a large difference in enzyme gene systems between strains. Could this be due to incomplete genome sequencing of the strains? It may be worth attempting to find answers through re-sequencing data.

(Response) Thank you for your valuable comment. It seems that the explanation in the manuscript and the presentation of the figure may not have fully conveyed all the necessary details. We have revised the descriptions of the CAZyme differences between the two strains and added three Supplementary Tables (Supplementary Table 7, 8, and 9).

We annotated the CAZymes in the genomes of Sanmaru 1-33, Sanjo 502-19, and B17 using dbCAN2. Among the total genes, 647 (3.9%) in B17, 656 (4.13%) in Sanmaru 1-33, and 685 (4.18%) in Sanjo 502-19 were annotated as CAZymes. The proportion of CAZyme genes relative to the total number of genes is very similar across the strains. Additionally, the number of genes in each CAZyme class is highly comparable (Supplementary Table 7). This suggests that gene prediction for each strain was performed adequately, allowing for a confident comparison of CAZyme class functions between Sanmaru 1-33 and Sanjo 502-19. Consequently, we identified CAZyme-annotated genes within strain-specific gene clusters based on orthologous gene cluster results for Sanmaru 1-33 and Sanjo 502-19 (Supplementary Tables 8 and 9). This enabled us to pinpoint functional differences specific to each strain. In summary, the overall gene analysis indicates that the differences in enzyme gene systems between the strains are not as drastic as initially thought. However, we were able to confirm strain-specific functional differences based on the orthologous gene cluster results.

# Revised:

  • We annotated the CAZymes in the genomes of Sanmaru 1-33, Sanjo 502-19, and B17 using dbCAN2 [47]. Among the total genes, 647 (3.9%) in B17, 656 (4.13%) in Sanmaru 1-33, and 685 (4.18%) in Sanjo 502-19 were annotated as CAZymes. The proportion of CAZyme genes relative to the total number of genes is very similar across the strains. Additionally, the number of genes in each CAZyme class is highly comparable (Supplementary Table 7). This suggests that gene prediction for each strain was performed adequately, allowing for a confident comparison of CAZyme class functions between Sanmaru 1-33 and Sanjo 502-19. Based on these results, we identified CAZyme-annotated genes within strain-specific gene clusters based on orthologous gene cluster analysis for Sanmaru 1-33 and Sanjo 502-19 (Supplementary Tables 8 and 9). This allowed us to identify functional differences specific to each strain. (Lines 325 - 335)
  • We added Supplementary Tables 7, 8, and 9 to provide detailed information on the strain-specific gene clusters of CAZyme-annotated genes.

(Reviewer’s comment #10) Besides the large VB segment identified on chromosome 9, are there any other segments or loci highly correlated with temperature sensitivity?

(Response) Thank you for your valuable comment. In this study, we focused on identifying large structural variants that may influence phenotypic changes. To explore genetic variations associated with high-temperature resistance (Sanmaru) and sensitivity (Sanjo), we investigated regions where variants such as SNVs occur in a structurally consecutive manner (regions with variations spanning over 50 kb) and where homozygous variants oppose each other for the two traits. From the genome-wide SNP data, we extracted genotypes in which the resistant samples had homozygous reference (REF) alleles, and the sensitive samples had homozygous alternative (ALT) alleles. Remarkably, this analysis revealed a large variant block (VB) in a continuous region of approximately 560 kb on chromosome 9, where the samples shared the same genotype. No other regions exhibited such continuous genotype blocks. We believe that this variant block is likely associated with the thermo-tolerant trait for fruiting body formation in Sanmaru compared to Sanjo. Furthermore, this genetic event supports the hypothesis that genetic recombination does not occur, thereby preserving high-temperature resistance in Sanmaru. We have added more details to the methods to better explain the approach used to identify variant blocks (VBs) from two distinct samples with high-temperature resistance and sensitivity (Line 361 – 367).

Regarding the area “highly correlated with temperature sensitivity,” we are planning to conduct follow-up research to investigate changes in gene expression within this region.

# Revised:

  • Based on the draft genome of Sanmaru, we analyzed single nucleotide variants (SNVs) between the two groups. We focused on regions where SNVs occur in a structurally consecutive manner (regions with variations spanning over 50 kb) and where homozygous variants oppose each other for the two traits. From the genome-wide SNP data, we extracted genotypes where the resistant samples had homozygous reference (REF) alleles, and the sensitive samples had homozygous alternative (ALT) alleles. (Line 361 – 367)

(Reviewer’s comment #11) The authors need to upload the gene annotation files for the two high-quality genomes completed in this paper to a public database, including gff, pep, cds, etc., rather than just uploading the genomic sequences.

(Response) Thank you for your valuable suggestion. While providing the data through a public database is indeed a good option, we believe it would be more beneficial to offer the gff, pep, and cds sequence files as supplementary materials through this journal. We have added the gene-model files for Sanmaru and Sanjo as requested in Supplementary Data: ‘Sanmaru.genemodel.gff-cds-pep.zip’ and ‘Sanjo.genemodel.gff-cds-pep.zip’. These gene model files are described in the Supplementary Materials.

Reviewer 2 Report

This article reports the genetic analysis of a genome-wide variant block linked to heat tolerance traits in Lentinula edodes. It contains important information to be published, but there are several points to be improved before publication.

1. In Figure 2D, glycoside hydrolase (GH) families containing potential β-glucosidases were more abundant in Sanjo than in Sanmaru, whether this phenomenon has potential significance?

2. In Figure 4B, please add DNA maker to the picture.

3. In this study, CAPS markers was used to distinguish proper temperature types. I have some small suggestions as to whether it is possible to combine transcriptome data to show more clearly the transcription levels of the relevant genes.

4. The numbering order of Supplementary Tables needs to be revised.

5. There are some errors in the reference information.

For example:

doi: ARTN 91025510.3389/fmicb.2022.910255.

doi: https://doi.org/10.1146/annurev.ge.22.120188.003215.

Author Response

Reviewer 2,

This article reports the genetic analysis of a genome-wide variant block linked to heat tolerance traits in Lentinula edodes. It contains important information to be published, but there are several points to be improved before publication.

(Reviewer’s comment #1) In Figure 2D, glycoside hydrolase (GH) families containing potential β-glucosidases were more abundant in Sanjo than in Sanmaru, whether this phenomenon has potential significance?

(Response) Thank you for your insightful question regarding the potential significance of the increased abundance of glycoside hydrolase (GH) families, including β-glucosidases. In our results (Figure 2D), genes categorized into the GH family were more abundant in Sanjo. A previous study by Li et al. (2021) suggested that GH families, which were analyzed from 21,244 bacterial, 424 archaeal, 456 viral, and 352 eukaryotic genomes in the CAZy database, have relatively narrow environmental distributions, with the highest abundance typically found in host-associated environments and a preference for moderate low-temperature and acidic environments. Moreover, they reported that GH subfamilies with a (β/α)8-barrel structure exhibit high resistance to high temperatures and are highly robust to mutations (Li et al. 2021). Remarkably, GHs with the (β/α)8-barrel structure, including AMY1, GUN4, and RHG4, were identified in our study, suggesting that these enzymes may enhance the functional diversity of Sanmaru at high temperatures. Additionally, the robustness to mutations observed in these GH subfamilies may be linked to the occurrence of a 560 kb variant block (VB) between Sanmaru and Sanjo. We have revised the description in the results section accordingly.

# Revised:

  • In this search, glycoside hydrolase (GH) families containing potential β-glucosidases were more abundant in Sanjo than in Sanmaru (Figure 2D). A previous study by Li et al. (2021) [48] suggested that GH families, which were analyzed from 21,244 bacterial, 424 archaeal, 456 viral, and 352 eukaryotic genomes in the CAZy database, have relatively narrow environmental distributions, with the highest abundance typically found in host-associated environments and a preference for moderate low-temperature and acidic environments. Moreover, they reported that GH subfamilies with a (β/α)8-barrel structure exhibit high resistance to high temperatures and are highly robust to mutations [48]. Remarkably, GHs with the (β/α)8-barrel structure, including AMY1, GUN4, and RHG4, were identified in our study, suggesting that these enzymes may enhance the functional diversity of Sanmaru at high temperatures. Additionally, the robustness to mutations observed in these GH subfamilies may be linked to the occurrence of a 560 kb variant block (VB) between Sanmaru and Sanjo. (Line 336 – 348).
  • A reference was added.

[48] Li DD; Wang JL; Liu Y; Li YZ; Zhang Z. Expanded analyses of the functional correlations within structural classifications of glycoside hydrolases. Comput Struct Biotechnol J. 2021;19:5931-5942. doi:10.1016/j.csbj.2021.10.039.

(Reviewer’s comment #2).  In Figure 4B, please add DNA maker to the picture.

(Response) We revised this comment.

(Reviewer’s comment #3).  In this study, CAPS markers was used to distinguish proper temperature types. I have some small suggestions as to whether it is possible to combine transcriptome data to show more clearly the transcription levels of the relevant genes.

(Response) Thank you for your valuable comments. The transcriptome analysis suggested by the reviewer was not possible to perform during the revision period and is planned as a follow-up study to this study and may be performed at a later date.

(Reviewer’s comment #4, 5).

 The numbering order of Supplementary Tables needs to be revised.

There are some errors in the reference information.

For example:

doi: ARTN 91025510.3389/fmicb.2022.910255.

doi: https://doi.org/10.1146/annurev.ge.22.120188.003215.

(Response) We are very sorry for these critical mistakes. We revised these comments.

Reviewer 3 Report

The authors have compared two phenotypes of Lentinula edodes: one that fruits at low temperature and one that fruits at high temperature. Then they looked for genetic differences across the genome to determine where these differ and also made markers to easily identify between the two phenotypes without needed to generate whole genomes. This is good because this fungus is a valuable edible fungus and being able to identify through a quick analysis if progeny or crosses will be able to fruit at higher temperatures will provide insight into which strains to keep for mass production.  Overall, the manuscript is well written and flows for the reader. To improve this manuscript, the authors should consider removing all the interpretative sentences within the results section (highlighted in the attached file) and move them to the discussion section and relate some of the authors findings to other research related to fungal variant blocks.  

Detained comments are within the attached file but in general, Figure 1 and Supplemental Figure 1 needs additional explanation for the reader to understand all aspects of the figure. For example, the green line and red arrow need to be explained in the figure legend. This is required for the reader to comprehend the elements within each figure. 

Within the results section: lines 229-230, 234-235, 269-270, 300-301, 304-305, 314-315, 330-333, 338-339, 362-364, are interpretation of the results and should be moved to the discussion section. These ideas should be incorporated within the discussion section and may provide the framework for a summary type conclusion paragraph.

Author Response

Reviewer 3,

The authors have compared two phenotypes of Lentinula edodes: one that fruits at low temperature and one that fruits at high temperature. Then they looked for genetic differences across the genome to determine where these differ and also made markers to easily identify between the two phenotypes without needed to generate whole genomes. This is good because this fungus is a valuable edible fungus and being able to identify through a quick analysis if progeny or crosses will be able to fruit at higher temperatures will provide insight into which strains to keep for mass production.  Overall, the manuscript is well written and flows for the reader. To improve this manuscript, the authors should consider removing all the interpretative sentences within the results section (highlighted in the attached file) and move them to the discussion section and relate some of the authors findings to other research related to fungal variant blocks.  

(Reviewer’s comment #1). Detained comments are within the attached file but in general, Figure 1 and Supplemental Figure 1 needs additional explanation for the reader to understand all aspects of the figure. For example, the green line and red arrow need to be explained in the figure legend. This is required for the reader to comprehend the elements within each figure. 

(Response) Thank you for this valuable comment. We revised this comment.

(Reviewer’s comment #2). Within the results section: lines 229-230, 234-235, 269-270, 300-301, 304-305, 314-315, 330-333, 338-339, 362-364, are interpretation of the results and should be moved to the discussion section. These ideas should be incorporated within the discussion section and may provide the framework for a summary type conclusion paragraph.

(Response) Thank you for your valuable feedback. We agree with your suggestion that the interpretative content within the Results section should be appropriately relocated to the Discussion section. However, a proper interpretation of the results can go a long way in helping the reader understand the essence of the experiment, so we have left some of the essential interpretation to the Results part.

Reviewer 4 Report

Overall, this is an excellent contribution, providing further understanding of the molecular mechanism behind high temperature tolerance for fungi, also good insights into the genetic variation for fruiting body production at high temperature for Shiitake mushroom. It has both good academic value and commercial potential. The manuscript is well prepared.

The submission creates association between genomic variations with thermo-tolerant fruiting body formation in Shitake mushroom (Lentinula edodes), the commercially important edible species. Inbred and hybrid populations were created for high temperature tolerant and sensitive cultivars. F1 populations were found high temperature tolerant and its F2 generation contains both high temperature tolerant and sensitive strains. Resequencing the F2 variants identified a large variant block (VB) of 560kb.  Within this VB, there is a significant enrichment of GO terms related to DNA damage stimulus and DNA repair. All the data provide strong support to the hypothesis that the genetic recombination was responsible for higher temperature resistance in Shiitake mushroom. CAPS markers were designed for discriminating thermotolerant strains successfully.

Author Response

Reviewer 4,

 Overall, this is an excellent contribution, providing further understanding of the molecular mechanism behind high temperature tolerance for fungi, also good insights into the genetic variation for fruiting body production at high temperature for Shiitake mushroom. It has both good academic value and commercial potential. The manuscript is well prepared.

The submission creates association between genomic variations with thermo-tolerant fruiting body formation in Shitake mushroom (Lentinula edodes), the commercially important edible species. Inbred and hybrid populations were created for high temperature tolerant and sensitive cultivars. F1 populations were found high temperature tolerant and its F2 generation contains both high temperature tolerant and sensitive strains. Resequencing the F2 variants identified a large variant block (VB) of 560kb.  Within this VB, there is a significant enrichment of GO terms related to DNA damage stimulus and DNA repair. All the data provide strong support to the hypothesis that the genetic recombination was responsible for higher temperature resistance in Shiitake mushroom. CAPS markers were designed for discriminating thermotolerant strains successfully.

(Response) Thank you for your kind comments. We appreciate the reviewers' efforts in improving our manuscript.